# The Role of Routine Electroencephalography in the Diagnosis of Seizures in Medical Intensive Care Units

**DOI:** 10.3390/diagnostics14111111

**Published:** 2024-05-27

**Authors:** Cheng-Lun Hsiao, Pei-Ya Chen, I-An Chen, Shinn-Kuang Lin

**Affiliations:** 1Stroke Center and Department of Neurology, Taipei Tzu Chi Hospital, Buddhist Tzu Chi Medical Foundation, New Taipei City 23142, Taiwan; shb@ms19.hinet.net (C.-L.H.); ruentw@gmail.com (P.-Y.C.); 2School of Medicine, Tzu Chi University, Hualien 97004, Taiwan; 3Taiwan Center for Drug Evaluation, Taipei 11557, Taiwan; iachen272@cde.org.tw

**Keywords:** critical illness, intensive care unit, routine electroencephalography, seizure, unfavorable outcome

## Abstract

Seizures should be diagnosed and treated to ensure optimal health outcomes in critically ill patients admitted in the medical intensive care unit (MICU). Continuous electroencephalography is still infrequently used in the MICU. We investigated the effectiveness of routine EEG (rEEG) in detecting seizures in the MICU. A total of 560 patients admitted to the MICU between October 2018 and March 2023 and who underwent rEEG were reviewed. Seizure-related rEEG constituted 47% of all rEEG studies. Totally, 39% of the patients experienced clinical seizures during hospitalization; among them, 48% experienced the seizure, and 13% experienced their first seizure after undergoing an rEEG study. Seventy-seven percent of the patients had unfavorable short-term outcomes. Patients with cardiovascular diseases were the most likely to have the suppression/burst suppression (SBS) EEG pattern and the highest mortality rate. The rhythmic and periodic patterns (RPPs) and electrographic seizure (ESz) EEG pattern were associated with seizures within 24 h after rEEG, which was also related to unfavorable outcomes. Significant predictors of death were age > 59 years, the male gender, the presence of cardiovascular disease, a Glasgow Coma Scale score ≤ 5, and the SBS EEG pattern, with a predictive performance of 0.737 for death. rEEG can help identify patients at higher risk of seizures. We recommend repeated rEEG in patients with ESz or RPP EEG patterns to enable a more effective monitoring of seizure activities.

## 1. Introduction

Electroencephalography (EEG) has become a standard diagnostic tool in clinical practice for analyzing electrical activity in the brain. EEG is also used in epilepsy research. Hans Berger was the first to successfully demonstrate neural oscillations in the human brain [1]. A routine EEG (rEEG) study usually involves obtaining a brief recording of brain activity and does not involve the video monitoring of the patient. However, technological advancements have led to the emergence of digital EEG, which has provided new development directions and enabled recordings of longer duration to be obtained [2]. Continuous EEG (cEEG) is a tool that continuously monitors brain function [3]. The clinical applications of rEEG and cEEG differ; rEEG studies can be performed in outpatient departments and in wards, whereas cEEG studies are usually only performed in wards or intensive care units (ICUs). cEEG studies are useful in the diagnosis of seizures, particularly nonconvulsive seizures (NCSs) and nonconvulsive status epilepticus (NCSE). cEEG studies are also useful for the prognostic evaluation of comatose patients in ICUs [4,5]. cEEG is more effective at detecting NCSs and NCSE than rEEG is; however, cEEG is not associated with improved outcomes compared with those of repeated rEEG [6,7,8].

Although cEEG studies are becoming more widely used in clinical settings, they are still infrequently used in ICUs, which is typically because ICUs lack the resources required for such studies and because of a general lack of evidence supporting improved outcomes from their use [3,9,10]. Tertiary medical centers often have stand-alone neurological ICUs that are equipped to perform cEEG studies, and critically ill patients without neurologically significant diseases are often admitted to the medical ICU (MICU). Neurologists often serve as consultants for patients admitted to the MICU for neurological concerns or symptoms. Although acute symptomatic seizures have been reported to occur in critically ill patients, seizures do not often occur in the MICU [11]. Seizures should be diagnosed and treated to ensure optimal health outcomes. EEG studies in the MICU may be arranged after consultation with a neurologist or upon the initiative of the attending physician. cEEG studies can aid in the diagnosis of epilepsy and improve a patient’s prognosis [4,5]. Several studies have investigated which patients in the MICU should be considered eligible for a cEEG study [12,13]. cEEG studies are not widely used, possibly because they are expensive and labor intensive. In addition, in 2021, the American Clinical Neurophysiology Society (ACNS) updated their guidelines regarding the standardization of critical care EEG terminology to prevent potential misinterpretation or overinterpretation [14,15,16]; however, the widespread uptake of standardized terminology requires a considerable amount of time.

In the present study, cEEG studies were not available in the MICU, and patients admitted to the MICU were only able to undergo rEEG studies. This study investigated the role of rEEG in the MICU (a ward not typically used to treat patients with neurological diseases). The study also explored whether rEEG is effective at detecting seizures and whether rEEG use was correlated with patient outcomes.

## 2. Materials and Methods

### 2.1. Study Population and Data Collection

This was a retrospective study involving patients who were admitted to the MICU of the study hospital and underwent rEEG between October 2018 and March 2023. Patients with overt epilepsy-based illnesses, such as convulsive status epilepticus or cluster seizures, who were admitted to the neurological ICU were excluded. The following information was collected: age; sex; reasons for admission to the MICU; indications for rEEG; Glascow Coma Scale (GCS) score during the rEEG study; whether a seizure during hospitalization in the MICU occurred before rEEG, within 24 h after rEEG, or at any time after EEG; and whether antiseizure medications (ASMs) were prescribed following rEEG. All the rEEG studies were performed in the MICU. Patients might undergo repeated rEEG studies during their stay in the MICU. We collected the first rEEG study from each patient for review. rEEG studies were arranged under the request of the MICU attending physicians, with or without a prior neurological liaison, to evaluate unexplained disturbed consciousness, detect seizure, or investigate prognosis. During the same period of study, a total of 410 rEEG studies were performed in the neurological ICU, which was fewer than those performed in the MICU.

### 2.2. Ethics Statement

The study was conducted in accordance with the Declaration of Helsinki. Ethical approval was provided by the Institutional Review Board of Taipei Tzu Chi Hospital, New Taipei City (approval no. 13-IRB034). The requirement for informed written consent was waived because the study was a retrospective data analysis. All data collected and analyzed in this retrospective study were derived from clinical records without any intervention or influence on clinical treatment. To ensure full protection of patient privacy and rights, only clinical observational data have been published; no personal information was disclosed to any third party without patient consent.

### 2.3. Reasons for Admission to the MICU

The etiologies of illnesses resulting in admission to the MICU were stratified into four main categories: respiratory failure, cardiovascular disease, sepsis or septic shock, and other disorders. Respiratory failure was considered to be hypoxia or hypercapnia requiring endotracheal intubation. Cardiovascular disease was considered to be cardiac arrest, acute myocardial infarction, myocarditis, and other heart disease. Sepsis or septic shock was considered to be any type of infection causing a life-threatening organ dysfunction. Other disorders included severe metabolic disorders or organ dysfunction.

### 2.4. Indications for rEEG

Indications for rEEG were classified as seizure related or non-seizure related. An EEG study was classified as seizure related if any seizure was observed during hospitalization in the emergency department, general ward, or ICU; if the physician reasonably suspected the possibility of seizure (i.e., if the physician observed fine involuntary movement of the limbs or the face); or if the patient had a history of epilepsy. An EEG study was classified as non-seizure related if the study was ordered for an unexplained disturbance of consciousness or for a prognostic evaluation.

### 2.5. Interpretation of EEG Patterns

All rEEG studies were recorded digitally by experienced technologists using a Nicolet vEEG system (VIASYS Healthcare, Middleton, WI, TSA) with 21 electrodes placed according to the International 10–20 system for scalp electrode placement and following ACNS guidelines, with a sampling rate of 250 Hz. The minimal recording duration was 15 min. All rEEG interpretations used standard 10–20 longitudinal bipolar recording as the primary montage. The results of each rEEG study were reported by general neurologists soon after the examination. We invited an experienced neurologist specialized in epilepsy and familiar with the latest version of the ACNS standardized critical care EEG terminology to conduct a retrospective review of each of the rEEG studies [16].

rEEG patterns were categorized into five groups: nonspecific slow waves (NSWs), suppression or burst suppression (SBS), sporadic epileptiform discharges (SED), rhythmic and periodic patterns (RPPs), and electrographic seizures (ESz; Figure 1).
**Nonspecific slow wave:** NSW indicated that no epileptic discharge met the ACNS requirements throughout the entire EEG recording except for slow waves and that the duration of suppression (amplitude < 10 μV) or attenuation (amplitude > 10 μV but less than half of the background amplitude) was less than 50% of the entire EEG recording duration.**Suppression or burst suppression:** We combined suppression and burst suppression/attenuation into a single category. The duration of suppression or attenuation was required to account for at least 50% of the total recording time. The burst duration was required to be longer than 0.5 s and for more than three phases [16]. This pattern usually indicates severe cortical dysfunction, which can result from being administered general anesthesia, having hypothermia or diffuse cortical damage, or being in a deep coma.**Sporadic epileptiform discharges:** Spikes, polyspikes, or sharp waves that met the definitions given by the ACNS but did not meet the definition of RPPs, ESz, or electroclinical seizures were interpreted as sporadic discharges regardless of EEG background [16]. These EEG findings often indicate cortical excitability, although the correlation between these EEG findings and epileptic seizures in unconscious patients is somewhat weak.**Rhythmic and periodic patterns:** Periodic discharges, rhythmic delta activity, and spike-and-wave or sharp-and-wave were all classified as RPPs and divided into generalized, lateralized, bilateral independent, unilateral independent, and multifocal patterns on the basis of where they occurred [16]. This type of EEG pattern has different correlations with epileptic seizures depending on where the pattern is detected, the morphology and frequency of the pattern, and whether the signal has plus modifiers (i.e., rhythmic, fast, or sharp waves superimposed on the RPP pattern). Lateralization, periodic discharges, higher frequency, and the combination of plus modifiers exhibit a higher correlation with epileptic seizures.**Electrographic seizures:** This type of epileptic seizure was diagnosed if epileptiform discharges averaging >2.5 Hz for ≥10 s or any pattern with the same definite evolution and lasting ≥10 s were detected. ESz lasting for ≥10 continuous min or for a total duration of ≥20% of any 60 min period of recording were defined as electrographic-status epilepticus and classified as ESz in our analysis [16].

### 2.6. Outcome Evaluation

The short-term outcome was assessed using the Glasgow Outcome Scale (GOS) at the time of discharge from the MICU. The GOS is used to assess functional outcomes on a 5-level scale with points 1 (death), 2 (vegetative status), 3 (severe disability), 4 (moderate disability), and 5 (good recovery). In the current study, a GOS score of ≤3 represented an unfavorable outcome.

### 2.7. Statistical Analysis

Because the current study’s continuous variables had a skewed distribution, medians alongside the 25th and 75th percentiles were used to explain the range. Mann–Whitney *U* or Kruskal–Wallis tests were used to evaluate differences in continuous variables. The chi-square or Fisher’s exact test was used for comparisons of categorical variables. Correlations between age and GOS scores or between GCS and GOS scores were analyzed using Spearman’s coefficient of rank correlation analysis. The continuous variables age and GCS score were converted into dichotomous variables using optimal cutoff values and determined in accordance with the Youden index with a receiver operating characteristic curve plotted for unfavorable outcomes and death prediction. Significant variables were included in a multiple logistic regression model to identify the factors significantly associated with an unfavorable or fatal outcome. Additionally, we analyzed the predictive performance of the variables through the concordance statistic (C-statistic) for a fatal outcome. A C-statistic, or the area under the receiver operating characteristic curve, used to assess the performance of predictive models particularly in binary classification tasks ranges from 0.5 (model performs no better than random chance) to 1.0 (model perfectly separates the two classes without any errors). A *p*-value of <0.05 was considered to indicate significance. Most of the statistical analyses were performed using SPSS (version 24; IBM, Armonk NY, USA). Scatter plot diagrams were constructed, and the correlations were compared using MedCalc (version 18; MedCalc Software, Mariakerke, Belgium).

## 3. Results

Over the study’s 5-year period, 560 eligible patients underwent 670 rEEG studies and were included in our analysis. Of the included patients, 91 (16%) underwent two or more rEEG studies (repeated rEEG; twice in 68 patients, three times in 19, four times in 2, and five times in 2). Table 1 lists the clinical features of the 560 patients. The median age was 74 years, and 281 patients (50%) were male. The median GCS score during rEEG was 8. Seizure-related rEEG studies comprised 47% of all EEG studies.

A clinical seizure, defined as any seizure that occurred during the entire course of hospitalization before discharge from the MICU, was observed in 219 patients (39%). Among these patients, 190 experienced a seizure before undergoing an rEEG study, and 29 experienced their first seizure after undergoing an rEEG study. In total, 76 patients experienced a seizure both before and after undergoing an rEEG study. A seizure was observed after rEEG and before discharge from the MICU in 105 patients (19%). Among these patients, 23 experienced a seizure 24 h after rEEG, with 3 developing NCSs. A seizure was observed within 24 h after rEEG in 82 patients (15%). Among these patients, 76 experienced a convulsive seizure, and 6 experienced an NCS. An NCS was observed in 22 (10%) of 219 patients with clinical seizure. Post-rEEG ASMs were prescribed for 290 patients (52%). Of the 219 patients who experienced a clinical seizure, 194 (89%) were prescribed post-rEEG ASMs. An unfavorable outcome (GOS score ≤ 3) occurred in 430 patients (77%), and 228 patients (41%) died. Compared with the patients without clinical seizures, those with clinical seizures were older, had more RPP (29%) and ESz (9%) EEG patterns, and were more likely to be prescribed post-rEEG ASMs. Among the 370 patients without seizures before rEEG, only 2 of 29 patients (7%) experiencing clinical seizures after rEEG had an SBS EEG pattern (*p* < 0.0001). Furthermore, of the 16 patients who experienced clinical seizures within 24 h after rEEG, none had an SBS EEG pattern (*p* < 0.0001).

The clinical features of patients with different medical illnesses are summarized in Table 2. The patients with cardiovascular diseases had the worst GCS scores (median score = 4) and outcomes (median GOS = 1) and had the highest mortality rate (64%). An NSW was the most frequently observed rEEG pattern in all disease groups (64–69%), with the exception of the cardiovascular disease group, for which SBS constituted the majority of the observed rEEG patterns (35%).

The clinical features of the patients with different rEEG patterns are summarized in Table 3. SED, RPPs, and ESz occurred in 50% or more of the patients who underwent seizure-related rEEG studies, with the RPP group being the most likely to undergo seizure-related rEEG studies (69%). Seizures within 24 h or during the entire course of the ICU stay after rEEG were observed most frequently in the ESz group (73% and 82%, respectively), second-most frequently in the RPP group (40% and 47%, respectively), and least frequently in the NSW group (3% and 6%, respectively). Repeated rEEG studies were performed most frequently in patients with an ESz rEEG pattern (64%), followed by patients with an RPP rEEG pattern (43%). The patients with an SBS rEEG pattern had the worst outcomes, with a median GOS score of 1. Unfavorable outcomes occurred in 90% or more of the patients with SBS (96%) and RPP (91%) rEEG patterns. Mortality was highest in the patients with an SBS rEEG pattern (81%).

Of 219 patients with clinical seizures, 105 (48%) experienced a seizure after their first rEEG study. Among these patients, 78% experienced a seizure within 24 h after rEEG. The clinical features of the patients who did and did not experience a post-rEEG seizure are summarized in Table 4. Compared with the patients without a post-rEEG seizure, those with one had lower GCS scores, more RPP and ESz EEG patterns, lower GOS scores, and a higher rate of unfavorable outcomes. Approximately half of the patients with a post-rEEG seizure underwent repeated rEEG studies. Mortality was significantly higher in the patients with a seizure within 24 h after rEEG (51% vs. 39%) but not significant in the patients with a seizure after rEEG. This finding suggests that the patients with a seizure within 24 h after rEEG had more serious clinical conditions. A post-rEEG seizure occurred in 91 patients (34%) of the 265 who underwent seizure-related rEEG studies and in 14 patients (5%) of the 295 who underwent non-seizure-related rEEG studies. Post-rEEG ASMs were prescribed to 95% of 105 patients with a post-rEEG seizure and to 98% of 82 patients with a seizure within 24 h after rEEG.

More than three-quarters (77%) of the patients experienced unfavorable short-term outcomes (GOS score ≤ 3 at the time of discharge from the MICU). Approximately 41% of patients had a fatal short-term outcome (GCS score = 1). The Spearman’s coefficient of the rank correlation analysis revealed a significant inverse correlation between the GOS score and age and a significant positive correlation between GOS and GCS scores (Figure 2). The ages of >68 and >59 years were the optimal cutoff values for the prediction of unfavorable outcomes and death, respectively, as determined using the Youden index with a receiver operating characteristic curve. The GCS scores of ≤9 and of ≤5 were the optimal cutoff values for the prediction of unfavorable outcomes and death, respectively. No correlation was observed between the presence of clinical seizures and unfavorable outcomes (Table 1). Nevertheless, the rate of post-rEEG seizures (including seizures within 24 h after rEEG) was significantly higher in patients with unfavorable outcomes. Given that the effect on the outcomes was similar between seizures within 24 h after rEEG (odds ratio: 5.453, *p* = 0.0003) and post-rEEG seizures (odds ratio: 5.187, *p* < 0.0001), we selected seizures within 24 h after rEEG for a further multivariable analysis. The multivariable analysis of the factors influencing short-term outcomes is presented in Table 5. Significant factors for unfavorable outcomes were age > 68 years, the presence of sepsis, respiratory failure, cardiovascular disease, a GCS score ≤ 9, SBS or RPP EEG patterns, and a seizure within 24 h after rEEG. Significant factors for death were age > 59 years, the male gender, the presence of cardiovascular diseases, a GCS score ≤ 5, and SBS EEG patterns. A C-statistic analysis of a fit model comprising these five significant predictors to estimate the predictive performance of death yielded a value of 0.737 (95% confidence interval: 0.694–0.779) (Figure 3).

## 4. Discussion

Seizure-related rEEG studies constituted approximately half (47%) of all rEEG studies included in our analysis. Additionally, 39% of the patients who underwent EEG studies experienced clinical seizures. Approximately half (48%) of the patients who experienced a clinical seizure experienced the seizure after undergoing an rEEG study, and 13% of the patients with clinical seizures experienced their first seizure after undergoing an rEEG study. The patients with a cardiovascular disease were the most likely to have the SBS EEG pattern and the highest mortality rate. The RPP and ESz EEG patterns were associated with a seizure within 24 h after rEEG, which was also related to unfavorable outcomes. More than three-quarters (77%) of the patients had unfavorable short-term outcomes. The significant predictors of death were age > 59 years, the male gender, the presence of cardiovascular disease, a GCS score ≤ 5, and the SBS EEG pattern.

EEG is a key tool in the diagnosis and treatment of epilepsy [17] and a valuable tool for early prognosis in comatose patients with posthypoxic encephalopathy [18]. However, no gold standard has been established for the indications of EEG studies. Both rEEG and cEEG are recommended for patients with generalized convulsive status epilepticus and for excluding NCSE in patients with unexplained and persistent altered consciousness. EEG is also useful for detecting ischemia in comatose patients with a subarachnoid hemorrhage and for improving the prognostication of comas [12,13]. cEEG is useful for identifying ischemia in patients at risk of cerebral ischemia, for assessing the level of consciousness in patients receiving intravenous sedation or a medically induced coma, and for assessing prognosis after cardiac arrest [13].

NCSE has consistently been considered a condition worthy of particular attention by physicians caring for comatose or critically ill patients. However, the precise incidence of NCSE is unknown. In the United States, the estimated incidence of status epilepticus is 15–20/100,000 per year, with as many as 63% of these cases potentially being NCSE [19]. Privitera et al. subjected patients with altered consciousness and without clinical convulsions to rEEG and found that 37% of patients had either defined or probable NCSE [20]. Towne et al. reviewed recorded cEEGs with at least 30 min of recording time and found that 8% of comatose patients in the ICU had NCSE [21]. Rudin et al. used rEEG to monitor patients during at least the first 30 min of treatment initiation and found that 81% of the cases of status epilepticus in the ICU were clinically nonconvulsive (47%) or subtle status epilepticus (34%) [22]. Most previous studies analyzed all EEG studies in the hospital or all ICU patients and did not analyze the MICU separately. In the current study, the incidence of NCSE (EEG pattern ESz) among the patients in the MICU who underwent rEEG studies was only 4%. This relatively low incidence is possibly due to the exclusion of patients with neurological crises or subtle status epilepticus and a limitation of seizure detection by rEEG. cEEG has been reported to be more effective at detecting epilepsy, particularly NCSE, than rEEG is [8,23]. Kamel et al. used cEEG to monitor patients in medical and surgical ICUs (excluding patients with acute brain injury, consistent with the exclusion criteria of the present study) and identified ESz in 11% of the patients [5]. Wagner et al. retrospectively reviewed 26,370 patients in ICUs under routine clinical practice. On the basis of a combination of rEEG and cEEG studies in patients with clinically unexplained impaired consciousness or altered neurologic function, they found that 200 patients (0.8%) experienced seizures. Among those patients, 20% did not have motor symptoms [11]. They addressed the limitation that certain NCSs may have been missed and the actual proportion of single or recurrent seizures may be higher than reported. An intensivist’s clinical observation and judgment play a crucial role in the decision to schedule rEEG studies, and these decisions can be made with or without neurological liaisons. In the present study, 14 patients who underwent non-seizure-related rEEG experienced a post-rEEG seizure. Among those patients, seven experienced a seizure within 24 h after rEEG, and five had NCSE (rEEG ESz pattern). Due to the lack of extensive rEEG studies, more patients with NCSE could be underdiagnosed. In our study, 64% of the patients with an ESz rEEG pattern underwent repeated rEEG. This suggests that repeated rEEG can help with establishing the diagnosis of NCSE and evaluating the effectiveness of treatment.

The SBS EEG pattern has long been observed in the human brain during general anesthesia, hypothermia, coma, severe brain injury, and early infantile encephalopathy [24]. Several studies have demonstrated an association of prolonged suppression or burst suppression with poor outcomes in patients with a postanoxic coma after cardiac arrest [25,26]. Clinical seizures were reported to occur in 29% of unconscious patients following cardiac arrest, indicating a poor outcome [27]. In addition, epileptiform EEG, occurring in approximately one-third of patients following cardiac arrest, also indicates a poor prognosis [28]. In the present study, most patients classified as having a cardiovascular disease had experienced cardiac arrest; 40% of the patients in this group experienced a seizure. The rate at which the SBS rEEG pattern was observed was highest among patients with cardiovascular diseases. The patients with cardiovascular diseases and those with an SBS rEEG pattern had the highest rates of an unfavorable outcome and death. Recently, the 6-point NEC2RAS score, requiring two rEEG or two 20 min cEEG clips at 12–36 h and 36–72 h after cardiac arrest, was established. This score was designed to predict independent life at 3 months with 100% sensitivity and 93% specificity [28].

Patients with a pattern of SEDs on early rEEG are more likely to subsequently develop RPPs and NCSs, and patients with RPPs on early rEEG are more likely to subsequently develop ESz [29,30]. This may indicate that SEDs, RPPs, and ESz are EEG manifestations of brain activities at different stages of progression from interictal to ictal. Therefore, RPPs are key indicators for seizure prediction. Higher frequencies of lateralized discharges (including bilateral independent), periodic discharges, and plus modifiers (rhythmic, fast, or sharp patterns superimposed on the signal) are considered to indicate a higher risk of epilepsy [31,32]. Seizure rates in the patients with RPPs ranged from 13% (generalized rhythmic delta activity) to 58% (lateralized periodic discharges) [32,33]. Koren et al. found that a combination of SED and RPPs was effective in predicting NCSE [30]. In the current study, we divided the EEG patterns into SEDs, RPPs, and ESz without the further analysis of the discharge frequencies or the subdivisions of RPPs. Struck et al. developed a simple point system (2HELPS2B) that incorporates six variables on the basis of cEEG findings and can be used to predict seizure risk in critically ill patients [33]. However, whether 2HELPS2B scores are also effective in predicting seizure risk on the basis of rEEG studies is not yet known.

Approximately 13–20% of critically ill patients undergoing cEEG have ESz. Most seizures are detected within 24 h after cEEG [34]. In our study, 105 patients experienced a seizure after rEEG, with 82 (78%) experiencing a seizure within 24 h after rEEG. Similarly, in a study by Zawar, 79.4% of patients experienced their first seizure within 24 h of undergoing a cEEG [34]. The ESz rEEG pattern was most strongly correlated with subsequent seizure, followed by the RPP EEG pattern. Although rEEG is not as effective as cEEG at predicting seizures, we recommend repeated rEEG for patients at high risk of seizure, particular those with ESz and RPP EEG patterns.

ASMs are indicated for patients with SED, RPP, or ESz EEG patterns. However, not all patients with the aforementioned rEEG patterns received post-rEEG ASMs in our study. The most common reason patients were not prescribed ASMs was the prioritization of correction of metabolic or infectious problems. In addition, some patients had a rapid deterioration to a fatal outcome before receiving ASMs, and diagnosis was delayed in a small number of patients due to imprecision in the initial interpretation of EEG patterns by general neurologists. Nevertheless, prompt and adequate ASM treatment may reduce the occurrence of unfavorable outcomes.

Neuroimaging studies can provide information about structural brain damage, and EEG can detect functional deficits in the brain [35]. Early EEG changes have been reported to be associated with increased mortality and the development of delirium in patients with sepsis [36]. Our study indicated that SBS and RPP rEEG patterns and seizures within 24 h after rEEG were significant risk factors for short-term unfavorable outcomes. Of these risk factors, only the SBS rEEG pattern was a significant predictor of death. The overall mortality rate in this study was 41%, and the significant predictors of death were age > 59 years, male gender, the presence of a cardiovascular disease, a GCS score ≤ 5, and the SBS rEEG pattern, with these factors having a predictive performance for mortality of 0.737. An assessment of brain function with EEG is as crucial as monitoring cardiorespiratory function is in critically ill patients.

The present study has several limitations. First, only patients who underwent rEEG were enrolled. We were not able to determine the proportion of patients who experienced a seizure among all MICU patients. Second, classifications of EEG patterns were retrospectively interpreted in accordance with the latest version of the ACNS guidelines. We did not compare these retrospective rEEG findings with the initially reported findings that were used to guide treatment after rEEG. This could explain why some of the patients with RPP or NCS EEG patterns did not receive post-EEG ASMs. Third, given that cEEG was not available at the study location, we were unable to compare the seizure detection rates between rEEG and cEEG. Hospitals do not always offer cEEG in the MICU. The benefits of rEEG highlighted in the present study have important clinical implications. Fourth, only patients admitted in the MICU were enrolled in this study. The reasons for ICU admission, indications for rEEG study, and distribution of rEEG patterns can be different in patients admitted to the neurological ICU. Furthermore, the effect of anesthesia used in patents with mechanical ventilation or seizure control was not analyzed. Finally, the follow-up period ended at the time of discharge from the MICU and not at the time of discharge from the hospital. All patients were admitted for critical illnesses; therefore, complex comorbidities may have affected final discharge outcomes. Furthermore, a GOS assessment is not a routine prognostic assessment performed upon patient discharge from a medical ward.

## 5. Conclusions

Clinical seizures occur more frequently in the MICU than in general wards. The careful observation of a patient’s subtle movements is required to detect NCSs when the patient has unexplained disturbed consciousness. In the absence of cEEG, rEEG can help detect abnormal brain activities and improve diagnostic accuracy and thereby lead to a more appropriate treatment. We recommend repeated rEEG in patients with ESz or RPP EEG patterns to enable the more effective monitoring of seizure activities.

## Figures and Tables

**Figure 1 diagnostics-14-01111-f001:**
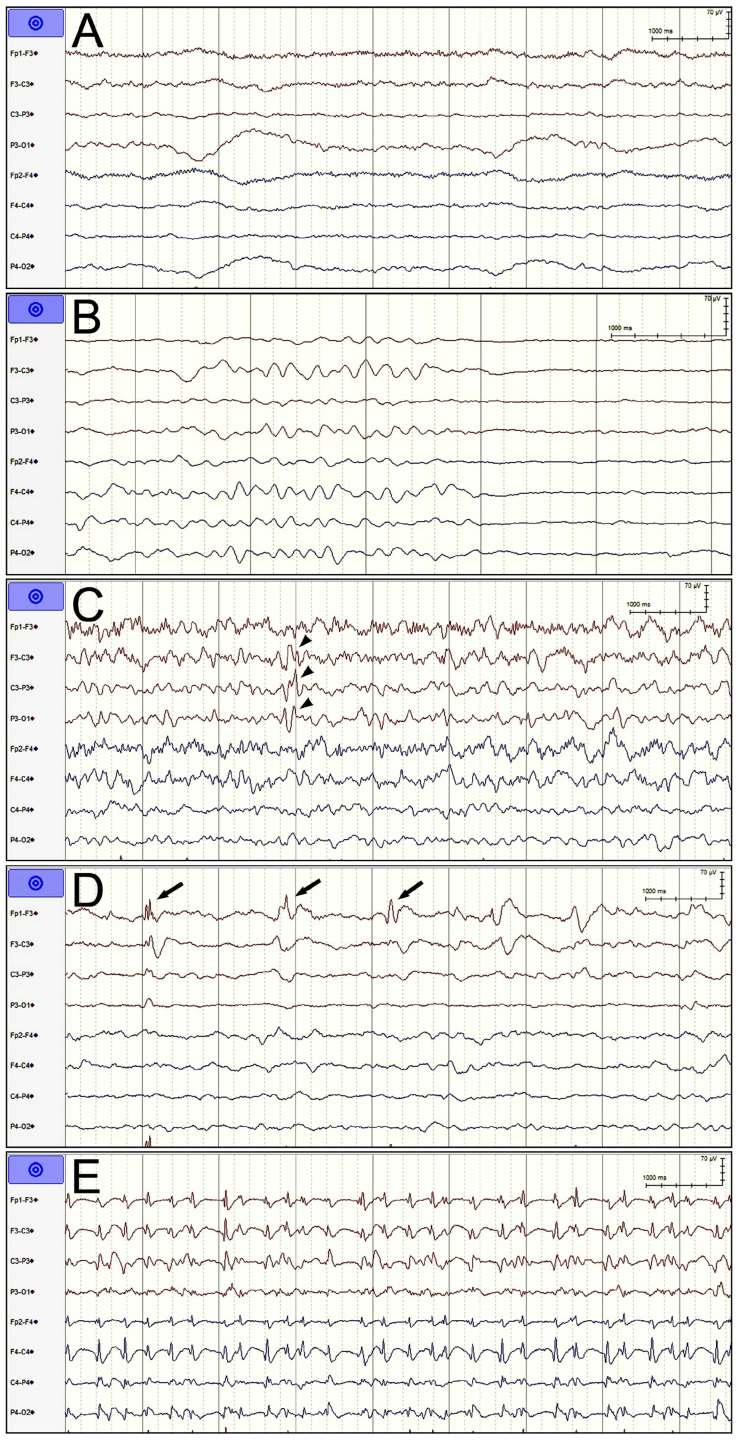
Five patterns of routine electroencephalography (displayed on bipolar montages). (**A**) Nonspecific slow waves, (**B**) suppression or burst suppression, (**C**) sporadic epileptiform discharges (arrowheads), (**D**) rhythmic and periodic patterns–lateralized sporadic discharges (arrows), and (**E**) electrographic seizures—3-Hz spike-and-wave complexes.

**Figure 2 diagnostics-14-01111-f002:**
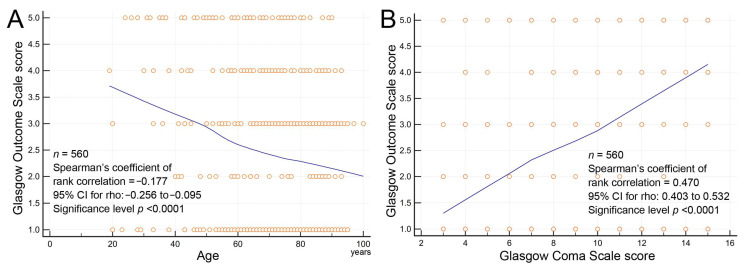
Spearman’s rank correlation coefficient plots revealing (**A**) strong inverse correlation between Glasgow Outcome Scale score and age and (**B**) strong positive correlation between Glasgow Outcome Scale score and Glasgow Coma Scale score.

**Figure 3 diagnostics-14-01111-f003:**
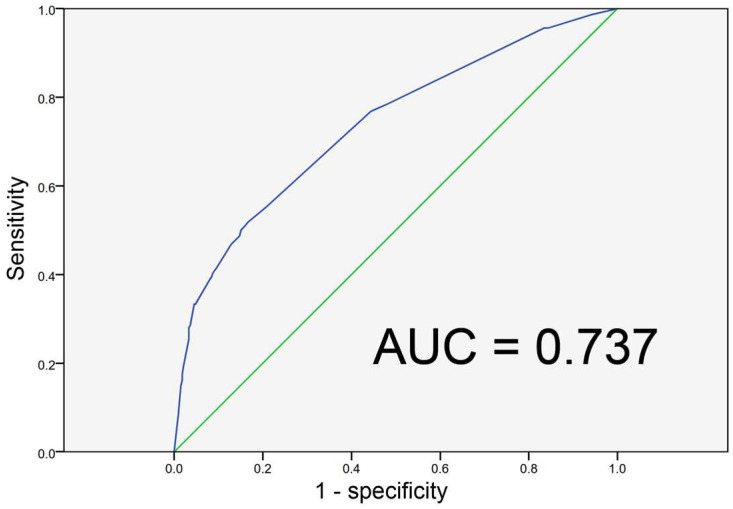
A C-statistic of 0.737 is estimated for the prediction of death from a fit model of the five significant predictors obtained from the regression analysis in Table 5. AUC, area under the curve.

**Table 1 diagnostics-14-01111-t001:** Clinical features of 560 critically ill patients who underwent electroencephalography.

Characteristic	Total (*n* = 560)	Clinical Seizure		*p* Value
		Yes (*n* = 219)	No (*n* = 341)	
Age (years)	74 (63–84)	71 (60–81)	77 (64–85)	0.0003
Gender (Male)	281 (50%)	110 (50%)	171 (50%)	>0.9999
GCS score	8 (5–10)	8 (4–10)	8 (5–10)	0.4933
Etiology of illness				0.2676
Respiratory failure	185 (33%)	63 (29%)	122 (36%)	
Cardiogenic diseases	139 (25%)	56 (26%)	83 (24%)	
Sepsis/septic shock	126 (23%)	50 (23%)	76 (22%)	
Other disorders	110 (20%)	50 (237%)	60 (1927%)	
Seizure-related rEEG	265 (47%)	205 (94%)	60 (18%)	<0.0001
rEEG patterns				<0.0001
NSW	325 (58%)	90 (41%)	235 (69%)	
SBS	69 (12%)	20 (9%)	49 (14%)	
SED	54 (10%)	27 (12%)	27 (8%)	
RPPs	90 (16%)	63 (29%)	27 (8%)	
ESz	22 (4%)	19 (9%)	3 (1%)	
Seizure within 24 h after rEEG	82 (15%)	82 (37%)	0 (0%)	<0.0001
Seizure after rEEG	105 (19%)	105 (48%)	0 (0%)	<0.0001
Post-rEEG ASM	290 (52%)	194 (89%)	96 (28%)	<0.0001
Repeated rEEG	91 (16%)	64 (29%)	27 (8%)	<0.0001
GOS score	3 (1–3)	3 (1–3)	3 (1–4)	0.3870
Unfavorable outcome	430 (77%)	175 (80%)	255 (75%)	0.1826
Death	228 (41%)	92 (42%)	136 (40%)	0.6596

ASM, antiseizure medications; ESz, electrographic seizures; GCS, Glasgow Coma Scale; GOS, Glasgow Outcome Scale; NSW, nonspecific slow wave; rEEG, routine electroencephalography; RPPs, rhythmic and periodic patterns; SBS, suppression or burst suppression; SED, sporadic epileptiform discharges.

**Table 2 diagnostics-14-01111-t002:** Clinical features of patients with various medical illnesses.

EEG Patterns	Respiratory Failure (*n* = 185)	Cardiovascular Diseases (*n* = 139)	Sepsis/Septic Shock (*n* = 126)	Other Disorders (*n* = 110)	*p* Value
Age (years)	78 (67–86)	71 (60–83)	74 (66–83)	68 (57–82)	0.0012
Male gender	86 (56%)	73 (53%)	64 (51%)	58 (53%)	0.6553
GCS score	8 (6–10)	4 (3–7)	8 (7–10)	10 (8–14)	<0.0001
rEEG patterns					<0.0001
NSW	120 (65%)	48 (35%)	81 (64%)	76 (69%)	
SBS	10 (5%)	49 (35%)	10 (8%)	0 (0%)	
SED	18 (10%)	10 (7%)	15 (12%)	11 (10%)	
RPPs	32 (17%)	24 (17%)	15 (12%)	19 (17%)	
ESz	5 (3%)	8 (6%)	5 (4%)	4 (4%)	
Seizure-related rEEG	91 (49%)	62 (45%)	58 (46%)	54 (49%)	0.8272
Clinical seizure	63 (34%)	56 (40%)	50 (40%)	50 (45%)	0.2671
Seizure within 24 h after rEEG	23 (12%)	27 (19%)	21 (17%)	11 (10%)	0.1342
Seizure after rEEG	32 (17%)	32 (23%)	27 (21%)	14 (13%)	0.1623
Post-rEEG ASM	88 (48%)	78 (56%)	67 (53%)	57 (52%)	0.4827
GOS score	3 (1–3)	1 (1–3)	3 (1–3)	3 (2–4)	<0.0001
Unfavorable outcome	152 (82%)	123 (88%)	97 (77%)	58 (53%)	<0.0001
Death	65 (35%)	89 (64%)	48 (38%)	26 (24%)	<0.0001

ASM, antiseizure medications; ESz, electrographic seizures; GCS, Glasgow Coma Scale; GOS, Glasgow Outcome Scale; NSW, nonspecific slow wave; rEEG, routine electroencephalography; RPPs, rhythmic and periodic patterns; SBS, suppression or burst suppression; SED, sporadic epileptiform discharges.

**Table 3 diagnostics-14-01111-t003:** Clinical features of patients with various rEEG patterns.

Characteristics	NSW (*n* = 325)	SBS (*n* = 69)	SED (*n* = 54)	RPPs (*n* = 90)	ESz (*n* = 22)	*p* Value
Age (years)	74 (63–84)	70 (61–83)	73 (58–85)	77 (66–86)	71 (63–83)	0.2507
Male gender	175 (54%)	36 (52%)	25 (46%)	34 (38%)	11 (50%)	0.1001
GCS score	9 (7–11)	3 (3–4)	9 (7–10)	7 (4–9)	5 (4–9)	<0.0001
Etiology of illness						<0.0001
Respiratory failure	120 (37%)	10 (14%)	18 (33%)	32 (36%)	5 (23%)	
Cardiovascular diseases	48 (15%)	49 (71%)	10 (19%)	24 (27%)	8 (36%)	
Sepsis/septic shock	81(25%)	10 (14%)	15 (28%)	15(17%)	5 (23%)	
Other disorders	76 (23%)	0 (0%)	11 (20%)	19 (21%)	4 (18%)	
Seizure-related rEEG	135 (42%)	24 (35%)	30 (56%)	62 (69%)	14 (64%)	<0.0001
Clinical seizure	90 (28%)	20 (29%)	27 (50%)	63 (70%)	19 (86%)	<0.0001
Seizure within 24 h after rEEG	10 (3%)	7 (10%)	13 (24%)	36 (40%)	16 (73%)	<0.0001
Seizure after rEEG	21 (6%)	9 (13%)	15 (28%)	42 (47%)	18 (82%)	<0.0001
Post-rEEG ASM	122 (36%)	34 (49%)	41 (76%)	76 (84%)	17 (77%)	<0.0001
Repeated rEEG	15 (5%)	9 (13%)	14 (26%)	39 (43%)	14 (64%)	<0.0001
GOS score	3 (1–4)	1 (1–1)	3 (1–3)	1 (1–3)	2 (1–3)	<0.0001
Unfavorable outcome	225 (69%)	66 (96%)	38 (70%)	82 (91%)	19 (86%)	<0.0001
Death	103 (32%)	56 (81%)	15 (28%)	46 (51%)	8 (36%)	<0.0001

ASM, antiseizure medications; ESz, electrographic seizures; GCS, Glasgow Coma Scale; GOS, Glasgow Outcome Scale; NSW, nonspecific slow wave; rEEG, routine electroencephalography; RPPs, rhythmic and periodic patterns; SBS, suppression or burst suppression; SED, sporadic epileptiform discharges.

**Table 4 diagnostics-14-01111-t004:** Clinical features of patients with and without post-rEEG seizure.

Characteristics	Seizure within 24 h after rEEG			Seizure after rEEG		
	Yes (*n* = 82)	No (*n* = 478)	*p* Value	Yes (*n* = 105)	No (*n* = 455)	*p* Value
Etiology of illness			0.1342			0.1623
Respiratory failure	23 (28%)	162 (34%)		32 (30%)	153 (34%)	
Cardiogenic diseases	27 (33%)	112 (23%)		32 (30%)	107 (24%)	
Sepsis/septic shock	21 (26%)	105 (22%)		27 (26%)	99 (22%)	
Other disorders	11 (13%)	99 (21%)		14 (13%)	96 (21%)	
GCS score	6 (3–8)	8 (6–10)	<0.0001	7 (4–9)	8 (5–10)	0.0002
Seizure-related rEEG	75 (91%)	190 (40%)	<0.0001	91 (87%)	174 (38%)	<0.0001
Post-rEEG ASM	80 (98%)	210 (44%)	0.0274	100 (95%)	190 (42%)	<0.0001
rEEG patterns			<0.0001			<0.0001
NSW	10 (12%)	315 (66%)		21 (20%)	304 (67%)	
SBS	7 (9%)	62 (13%)		9 (9%)	60 (13%)	
SED	13 (16%)	41 (9%)		15 (14%)	39 (9%)	
RPPs	36 (44%)	54 (11%)		42 (40%)	48 (11%)	
ESz	16 (20%)	6 (1%)		18 (17%)	4 (1%)	
Repeated rEEG	42 (51%)	49 (10%)	<0.0001	52 (50%)	39 (9%)	<0.0001
GOS score	1 (1–3)	3 (1–4)	0.0010	2 (1–3)	3 (1–4)	0.0008
Unfavorable outcome	77 (94%)	353 (74%)	<0.0001	98 (93%)	332 (73%)	<0.0001
Death	42 (51%)	186 (39%)	0.0361	50 (48%)	178 (39%)	0.1101

ASM, antiseizure medications; ESz, electrographic seizures; GCS, Glasgow Coma Scale; GOS, Glasgow Outcome Scale; NSW, nonspecific slow wave; rEEG, routine electroencephalography; RPPs, rhythmic and periodic patterns; SBS, suppression or burst suppression; SED, sporadic epileptiform discharges.

**Table 5 diagnostics-14-01111-t005:** Multivariable analysis of factors influencing short-term unfavorable outcomes.

Unfavorable Outcome (*n* = 430)			Death (*n* = 228)		
Characteristics	Odds Ratio (95% CI)	*p* Value	Characteristics	Odds Ratio (95% CI)	*p* Value
Age > 68 years	3.542 (2.172–5.776)	<0.0001	Age > 59 years	2.150 (1.291–3.581)	0.0032
Male gender	0.915 (0.565–1.482)	0.7180	Male gender	1.581 (1.072–2.331)	0.0207
Etiology of illness ^a^			Etiology of illness ^a^		
Sepsis/septic shock	1.986 (1.037–3.804)	0.0384	Sepsis/septic shock	1.572 (0.862–2.866)	0.1401
Respiratory failure	2.805 (1.529–5.146)	0.0009	Respiratory failure	1.335 (0.762–2.341)	0.3126
Cardiovascular diseases	3.005 (1.396–6.470)	0.0049	Cardiovascular diseases	2.447 (1.289–4.645)	0.0062
GCS ≤ 9	4.536 (2.795–7.359)	<0.0001	GCS ≤ 5	3.021 (1.827–4.993)	<0.0001
rEEG Pattern ^b^			rEEG Pattern ^b^		
SBS	4.000 (1.115–14.353)	0.0335	SBS	3.324 (1.555–7.107)	0.0019
SED	0.860 (0.413–1.789)	0.6859	SED	0.856 (0.433–1.693)	0.6551
RPPs	3.383 (1.360–8.416)	0.0087	RPPs	1.654 (0.940–2.911)	0.0810
ESz	1.183 (0.254–5.516)	0.8307	ESz	0.522 (0.172–1.583)	0.2505
Seizure within 24 h after rEEG	4.260 (1.335–13.593)	0.0143	Seizure within 24 h after rEEG	1.316 (0.701–2.470)	0.3928

^a^ Using other disorders as reference etiology; ^b^ Using nonspecific slow wave as reference pattern; CI, confidence interval; ESz, electrographic seizures; GCS, Glasgow Coma Scale; rEEG, routine electroencephalography; RPPs, rhythmic and periodic patterns; SBS, suppression or burst suppression; SED, sporadic epileptiform discharges.

## Data Availability

The data presented in this study are available on request from the corresponding author.

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
