# Peer review of "The Role of Routine Electroencephalography in the Diagnosis of Seizures in Medical Intensive Care Units"

_diagnostics, 2024, doi:10.3390/diagnostics14111111_

Round 1

Reviewer 1 Report

Comments and Suggestions for Authors

In this study, the authors investigated the effectiveness of routine EEG (rEEG) in 12 detecting seizures in the medical intensive care unit (MICU). A total of 560 rEEG were reviewed in a retrospective study, including a time range between 2018 and 2023 of hospitalization cases. Using a multiple logistic regression model, they identify among the factors significantly associated with unfavorable or fatal outcomes, a seizure within 24h after rEEG and SBG EEG pattern. They also performed a C-statistics on an AUC of the five significant model predictors with a score of 0.737, indicating moderate discriminative ability for the predictive model. The study is interesting, and it can have a considerable impact on the field. 

I only have a few remarks that I believe should be addressed before publication. 

Please provide more details about the EEG recording conditions, such as the number of electrodes, sampling frequency, the electrode of reference, and the type of montage used. From Figure 1, it appears to be a bipolar montage, but this is not explicitly stated. Additionally, the duration of the EEG recording is not specified. Is there a standard duration for your study? These parameters are important for the repeatability and comparison of your research with others and for the standardization of EEG recording practices in the field, which can significantly improve the quality and reliability of diagnosis outcomes. 

Please indicate if "(O.694-0.779)" in Line 281 refers to the CI. 

It would be beneficial to elaborate on the C-statistic analysis in the 2.7 paragraph. Since this statistic is known by different names, it would be helpful to specify that it refers to: "a Concordance statistic or the Area under the Receiver Operating Characteristic Curve (AUC-ROC), used to assess the performance of predictive models, particularly in binary classification tasks. The C-statistic ranges from 0.5 (model performs no better than random chance) to 1.0 (model perfectly separates the two classes without any errors)." 

Similarly, the Roc Curve reported in Figure 3 should be explained in more detail to emphasize the outcome's importance. 

Author Response

Manuscript ID diagnostics-2995647

Title: Role of routine electroencephalography in the diagnosis of seizure in medical intensive care units

Thanks to reviewer’s precious comments. We have checked the manuscript and have made essential revisions according to reviewer’s comments point-by-point. We have also corrected some typos in the text. The revised portions in the manuscript were coded in red color.

Comments from Reviewer 1:

  1. Please provide more details about the EEG recording conditions, such as the number of electrodes, sampling frequency, the electrode of reference, and the type of montage used. From Figure 1, it appears to be a bipolar montage, but this is not explicitly stated. Additionally, the duration of the EEG recording is not specified. Is there a standard duration for your study? These parameters are important for the repeatability and comparison of your research with others and for the standardization of EEG recording practices in the field, which can significantly improve the quality and reliability of diagnosis outcomes.

Response: We have added more details about the EEG recording conditions in Section 2.5 and more information about montage in the Figure 1 legends.

  1. Please indicate if "(0.694-0.779)" in Line 281 refers to the CI.

Response: We have added “95% confidence interval:” in the text.

  1. It would be beneficial to elaborate on the C-statistic analysis in the 2.7 paragraph. Since this statistic is known by different names, it would be helpful to specify that it refers to: "a Concordance statistic or the Area under the Receiver Operating Characteristic Curve (AUC-ROC), used to assess the performance of predictive models, particularly in binary classification tasks. The C-statistic ranges from 0.5 (model performs no better than random chance) to 1.0 (model perfectly separates the two classes without any errors)." Similarly, the ROC curve reported in Figure 3 should be explained in more detail to emphasize the outcome's importance.

Response: We have added more description about C-statistic in Section 2.7, and have modified the legends of Figure 3 to explain the area under the ROC curve of C-statistics.

Reviewer 2 Report

Comments and Suggestions for Authors

I reviewed carefully the manuscript titled: Role of Routine Electroencephalography in the Diagnosis of 2 Seizure in Medical Intensive Care Units, however, major and minor modifications are needed as follows:

Major

In the abstract add the years of inclusion of the patients in the study. In the abstract the assumption of better decision-making in treatment and outcome is beyond this study.

In the introduction only used the current ACNS terminology publication as current evidence if (lines 60-63), old information is not needed. Current EEG terminology is not standardized, 60 which can lead to potential misinterpretation or overinterpretation [3,9,14,15]. In 2021, the 61 American Clinical Neurophysiology Society (ACNS) updated their guidelines regarding 62 the standardization of critical care EEG terminology [16];

Methods 2.5 add the duration of your standard r-EEG recording without video or according to the ACNS guidelines added.

Were the EEG patterns defined in the Method section according to ACNS guidelines cEEG terminology? or are adjusted to only review of the interpreters?

Only one rEEG was considered to the analysis? Which one was selected the first in UCI or in other area? At what timing was performed the EEG?  How often rEEG are performed in your center? These data is needed to mention in the Methodological section.

How many interpreters were involved in the rEEG reports or study?

In the result section there a unneeded redundant information between tables and text.

In the Discussion, another limitation is the bias selection between neurological and not neurological ICU and the use of anesthesia specially in patients with mechanical ventilation.

In the Discussion section 317-323 is needed to mention the setting from these epidemiological studies to compared with non-neurological ICU units. Also, However, not 386 all patients with the rEEG patterns received post-rEEG ASMs in our 387 study need to be justified as acute symptomatic seizures that not neede ASMs.

In the conclusion, why recommended this Careful 420 observation of a patient’s subtle movements is required to detect seizures, particularly 421 NCSs, if this patients were excluded in this study.

Minor

Correct the grammatical issue: thrice

Comments on the Quality of English Language

mentioned in the previous section.

Author Response

Manuscript ID diagnostics-2995647

Title: Role of routine electroencephalography in the diagnosis of seizure in medical intensive care units

Thanks to reviewer’s precious comments. We have checked the manuscript and have made essential revisions according to reviewer’s comments point-by-point. We have also corrected some typos in the text. The revised portions in the manuscript were coded in red color.

Comments from Reviewer 1:

Major:

  1. In the abstract add the years of inclusion of the patients in the study. In the abstract the assumption of better decision-making in treatment and outcome is beyond this study.

Response: We have added the period of patient enrollment in [Abstract]. We have also revised the conclusion of [Abstract].

  1. In the introduction only used the current ACNS terminology publication as current evidence if (lines 60-63), old information is not needed. Current EEG terminology is not standardized, 60 which can lead to potential misinterpretation or overinterpretation [3,9,14,15]. In 2021, the 61 American Clinical Neurophysiology Society (ACNS) updated their guidelines regarding 62 the standardization of critical care EEG terminology [16].

Response: We have removed unnecessary description about old information in [Introduction].

  1. Methods 2.5 add the duration of your standard r-EEG recording without video or according to the ACNS guidelines added.

Response: We have added more detailed description about rEEG study in Section 2.5.

  1. Were the EEG patterns defined in the Method section according to ACNS guidelines cEEG terminology? or are adjusted to only review of the interpreters?

Response: The rEEG patterns defined in [Method] Section 2.5 were based on the ACNS standardized critical care EEG terminology. We added a pattern of nonspecific slow wave (NSW), which is not included in the ACNS terminology, for those without seizure activities.

  1. Only one rEEG was considered to the analysis? Which one was selected the first in UCI or in other area? At what timing was performed the EEG? How often rEEG are performed in your center? These data is needed to mention in the Methodological section.

Response: We have revised the statements in [Methods] Section 2.1: “All the rEEG studies were performed in the MICU. Patients might undergo repeated rEEG studies during their stay in the MICU. We collected the first rEEG study from each patient for review. rEEG studies were arranged under the request of the MICU attending physicians, with or without a prior neurological liaison, to evaluate unexplained disturbed consciousness, detect seizure, or investigate prognosis. During the same period of study, a total of 410 rEEG studies were performed in the neurological ICU, which was fewer than those performed in the MICU.”

  1. How many interpreters were involved in the rEEG reports or study?

Response: Only on doctor was involved in the interpretation of rEEG studies in this study. We have mentioned this information in [Methods] 2.5: “We invited an experienced neurologist specialized in epilepsy and familiar with the latest version of the ACNS standardized critical care EEG terminology to conduct a retrospective review of each of the rEEG studies.”

  1. In the result section there a unneeded redundant information between tables and text.

Response: We have deleted redundant information in the text in [Results].

  1. In the Discussion, another limitation is the bias selection between neurological and not neurological ICU and the use of anesthesia specially in patients with mechanical ventilation.

Response: We have added one more limitation regarding bias selection between neurological and medical ICUs, and the effect of anesthesia.

  1. In the Discussion section 317-323 is needed to mention the setting from these epidemiological studies to compared with non-neurological ICU units. Also, However, not 386 all patients with the rEEG patterns received post-rEEG ASMs in our 387 study need to be justified as acute symptomatic seizures that not neede ASMs.

Response: We have added “Most previous studies analyzed all EEG studies in the hospital, or all ICU patients, and did not analyze the MICU separately.” to explain the difference between our study and previous studies. We have also added “Nevertheless, prompt and adequate ASMs treatment may reduce the occurrence of unfavorable outcomes.” to emphasize the necessity of ASMs treatment.

  1. In the conclusion, why recommended this Careful 420 observation of a patient’s subtle movements is required to detect seizures, particularly 421 NCSs, if this patients were excluded in this study.

Response: Patients with NCSs were not excluded from this study. However, it is possible that more NCS patients admitted in the MICU were not correctly diagnosed due to a lack of EEG studies. We have modified the description as below: “Careful observation of a patient’s subtle movements is required to detect NCSs when the patient has unexplained disturbed consciousness.”

Minor:

  1. Correct the grammatical issue: thrice

Response: We have changed “thrice” to “three times”.

  1. Comments on the Quality of English Language mentioned in the previous section.

Response: The manuscript has been sent for a professional English editing with attached editorial proof before first submission. We have carefully rechecked the English grammar again before submitting the revised version.

Round 2

Reviewer 2 Report

Comments and Suggestions for Authors

None